



# Two-year intercomparison of three methods for measuring black carbon concentration at a high-altitude research station in Europe

Sarah Tinorua[1], Cyrielle Denjean[1], Pierre Nabat[1], Véronique Pont[2], Mathilde Arnaud[1],
Thierry Bourrianne[1], Maria Dias Alves[2], and Eric Gardrat[2]

[1]CNRM, Université de Toulouse, Météo-France, CNRS, Toulouse, France
[2]Laboratoire d'Aérologie, UPS Université Toulouse 3, CNRS (UMR 5560), Toulouse, France

**Correspondence:** Sarah Tinorua (sarah.tinorua@umr-cnrm.fr), Cyrielle Denjean (cyrielle.denjean@meteo.fr)

**Abstract.** Black carbon (BC) is one of the most important climate forcer with severe health effects. Large uncertainties in radiative forcing estimation and health impact assessment arise from the fact that there is no standardised method to measure BC mass concentration. This study presents a two-year comparison of three state-of-the-art BC measurement techniques at the high-altitude research station Pic du Midi located in the French Pyrenees at an altitude of 2877 m above sea level. A recently

upgraded aethalometer AE33, a thermal-optical analyzer Sunset and a single-particle soot photometer SP2 were deployed to measure simultaneously the mass concentration of equivalent black carbon ($M_{eBC}$), elemental carbon ($M_{EC}$) and refractory black carbon ($M_{rBC}$), respectively. Significant deviations in the response of the instruments were observed. All techniques responded to seasonal variations of the atmospheric changes in BC levels and exhibited good correlation during the whole study period. This indicates that the different instruments quantified the same particle type, despite the fact that they are based

on different physical principles. However the slopes and correlation coefficients varied between instrument pairs. The largest biases were observed for the AE33 with $M_{eBC}$ values that were around 2 times greater than $M_{rBC}$ and $M_{EC}$ values. The principal reasons of such large discrepancy was explained by the too low MAC and C values recommended by the AE33 manufacturer and applied to the absorption coefficients measured by the AE33. In addition, the long-range transport of dust particles at PDM in spring caused significant increases in the bias between AE33 and SP2 by up to a factor 8. The Sunset

$M_{EC}$ measurements agreed within around 17% with the SP2 $M_{rBC}$ values. The largest overestimations of $M_{EC}$ were observed when the total carbon concentration were below 25 $\mu gC\ cm^{-2}$, which is probably linked to the incorrect determination of the OC-EC split point. Another cause of the discrepancy between instruments was found to be the limited detection range of the SP2, which did not allow the total detection of fine rBC particles. The procedure used to estimate the missing mass fraction of rBC not covered by the measurement range of the SP2 was found to be critical. We found that a time-dependent correction

based on fitting the observed rBC size distribution with a multimodal lognormal distribution are needed to accurately estimate $M_{rBC}$ over a larger size range.



# 1 Introduction

Black carbon (BC), which results from incomplete combustion of fossil fuels, biofuel, and biomass, is one of the most important short-lived climate forcer (IPCC, 2022). Due to its strong absorption in the visible wavelengths, it can reduce the amount of sunlight reaching the surface, heat the atmospheric layer in which it resides and affects cloud formation, dissipation, precipitation with ensuing effects on atmosphere circulation through semi-direct radiative effects (Wang et al., 2016; Matsui et al., 2018; Tang et al., 2020). When deposited on the cryosphere (e.g. glaciers, snow cover, and sea ice), BC can reduce the surface albedo, thereby accelerating melt (Réveillet et al., 2022; Jacobi et al., 2015). Moreover, BC poses a threat to human health as it is considered as a carcinogen and source of respiratory disease due to its nanometer size (Janssen et al., 2012).

BC mass concentration ($M_{BC}$) data are required to develop, assess, and improve emission inventories, climate and chemical-transport model simulations, and mitigation strategies designed to both reducing air pollution and climate change. One major issue in $M_{BC}$ measurements is related to the lack of internationally accepted standardised method to measure it. Bond et al. (2013) discussed limitations in inferring its atmospheric concentration and highlighted inconsistencies between different terminology and related measurement techniques. Petzold et al. (2013) defined a specific nomenclature for BC according to its quantification method. Following the recommendation of the authors $M_{BC}$ can be categorised into three broad measurement techniques: (1) filter-based optical methods, which measure light attenuation and convert it to an equivalent BC mass concentration ($M_{eBC}$); (2) thermo-optical analysis methods, which report elemental carbon mass concentration ($M_{EC}$) as the mass concentration of carbon which is thermally refractory up to about 800 K (depending on the analysis protocol); and (3) laser-induced incandescence (LII) methods, which measure refractory BC mass concentration ($M_{rBC}$) as the incandescence signal of sampled particles after rapid heating to ∼4000K. Since there is not yet a universally accepted $M_{BC}$ quantification technique, it is extremely important to understand how the measurements vary between different instruments and techniques and what the reasons behind these potential differences are.

Filter-based optical methods are commonly used for $M_{eBC}$ measurements at long-term research sites such as the Global Atmosphere Watch (GAW), and the Aerosol, Clouds and Trace Gases Research Infrastructure (ACTRIS) programs because they are inexpensive and easy to maintain. Comparison of the different optical methods revealed discrepancies up to 45% among instruments of the same type (Arnott et al., 2006; Chow et al., 2009; Müller et al., 2011; Laing et al., 2020; Wu et al., 2015; Mason et al., 2018; Davies et al., 2018; Cuesta-Mosquera et al., 2021) and up to a factor of 5 when comparing thermal-optical and LII methods (Healy et al., 2017; Laing et al., 2020; Slowik et al., 2007; Chirico et al., 2010; Sharma et al., 2017). Quantifying $M_{eBC}$ acquired by optical methods is challenging because it requires the assumption of a mass absorption cross section (MAC) value translating the absorption coefficient ($\sigma_{abs}$). Field and laboratory measurements have indicated that MAC vary both temporally and spatially with values ranging from 3.8 to 58 m² g$^{-1}$(Wei et al., 2020). The wide range of reported values is not surprising given that the MAC relies on the BC core diameter, coating thickness, chemical composition and shape, which are expected to be influenced by a variety of spatio-temporal factors such as source type, transport pathway and regional atmospheric composition and meteorology. Still further complications arise from the fact that the optical methods are prone to several filter artifacts, including dependence of light attenuation on the filter tape loading, the interference of aerosol light





scattering with the absorption measurement and the multiple light scattering effects of the filter itself (Bond et al., 1999; Wein-gartner et al., 2003; Collaud Coen et al., 2010; Lack et al., 2014; Liousse et al., 1993; Schmid et al., 2006).

Thermal–optical and LII techniques are the most direct methods to measure $M_{BC}$. Both techniques make use of the high refrac-toriness of BC to quantify its mass concentrations, although in different manners. There is considerable variability in results
of field campaigns comparing $M_{EC}$ and $M_{rBC}$. Some studies have shown that $M_{rBC}$ measured by a SP2 and $M_{EC}$ measured by a Sunset analyzer were consistent within measurement uncertainties (Laborde et al., 2012a; Corbin and Gysel-Beer, 2019; Miyakawa et al., 2016). Other studies have shown they can systematically differ by factors of up to 2.5 (Pileci et al., 2021; Zhang et al., 2016; Sharma et al., 2017). While the authors could not clearly assign the reasons for the discrepancies to one or the other method, they found that various interferences from co-emitted species in the Sunset analyzer and the different particle
size range covered by the two methods could be the reason for the discrepancies.

Most instrument inter-comparisons took place in the planetary boundary layer (PBL), whereas very few intercomparisons at upper altitudes are available in literature. Laing et al. (2020) found that $M_{eBC}$ measured by an aethalometer at a US mountain site in the summer was 2 times higher than $M_{rBC}$ measured by a SP2 when using the aethalometer manufacter's recommen-dations for corrections. Observations in the free troposphere (FT) are more difficult to perform than at lower altitudes due
to the lack of availability of suitable sites and due to adverse meteorological conditions. Airborne studies can overcome such problems but are usually limited to short time scales, and thus do not provide statistically representative information at seasonal time scale. Moreover, distinguishing between signals, noise and inter-instrument uncertainty may become challenging at high altitude, as $M_{BC}$ can be several orders of magnitude lower in the FT than in the PBL (Sun et al., 2021). These aspects have historically kept intercomparison of BC measurements in the FT very sparse.
In this work, we conducted a systematic comparison of three current state-of-the-art BC-monitoring instruments at the high-altitude research site of Pic du Midi in the French Pyrenees (PDM, 2877 m asl.). More specifically, the recently upgraded aethalometer (AE33, Magee Scientific, Berkeley, CA), analyses of filter samplings with a thermal-optical analyzer (Sunset) and an online single-particle soot photometer (SP2) were deployed continuously during two years to measure $M_{eBC}$, $M_{EC}$ and $M_{rBC}$, respectively. The purpose of this study is to evaluate the agreement between the three measurement techniques, to
highlight the possible source of biases and to provide some recommendations on the use and data analysis of these different instruments.

## 2 Methods

### 2.1 Measurements site

The Pic du Midi research station (PDM, 42.93642°N, 0.14260°E) located in the South-West of France, is part of the Pyrenees
mountain chain, with an altitude of 2877 m asl. This site belongs to the European Aerosols, Clouds, and Trace gases Research InfraStructure (ACTRIS-Fr) and to the Global Atmosphere Watch (GAW) program of the World Meteorological Organisation. It is often under the FT influence with limited local pollution around the site (Collaud Coen et al., 2018; Gheusi et al., 2016; Tinorua et al., 2023). It is therefore considered as a background mountain site. Air masses arriving at the PDM have various





| Parameter | Instrument | Abbr. | Time Res. | Averag. Time | Measurement principle | Measurement uncertainty | Other notes |
|---|---|---|---|---|---|---|---|
| Refractory black carbon (rBC) | Single Particle Soot Photometer | SP2 | 1 sec | hourly | Laser induced incandescence of single particle | 24.5% (quadratic sum of sampling flow, anisokinetic sampling errors and $R_{fit/meas}$ factor errors) | Observed rBC mass distribution fitted by a daily multimodal lognormal size distribution |
| Equivalent black carbon (eBC) | 7-wavelength Aethalometer with Dual Spot$^{TM}$ technology | AE-33 | 2 min | hourly | Light absorption | 35% (Zanatta et al., 2016) | Applying filter type correction using C = 1.39 (filter M8060), MAC = 7.77 m²g$^{-1}$ |
| Elemental carbon (EC) | Semi-continuous carbon aerosol analysis | Sunset | 7 days | weekly | Light absorption and volatility | 16% (Liu et al., 2013) | Analysing using EUSAAR-2 temperature protocol |

**Table 1.** Summary of BC instruments and data analysis protocol used in this study

geographical origins coming from the Continental Europe, as well as over the Atlantic-Ocean, Iberian Peninsula and North
Africa. Therefore, the PDM is a suitable site to study BC long-range transport in the lower FT.

## 2.2   Instrumentation

From February 2019 to January 2021, an important set of instruments has been deployed to measure BC microphysical, chemical and optical properties in the framework of the h-BC project (Tinorua et al., 2023). Among them, three instruments were dedicated to the quantification of $M_{BC}$: a recently upgraded aethalometer (model AE33, Magee Scientific Company, Berkeley,
CA, USA), a thermal-optical analyzer (Sunset Laboratory Inc., Tigard, OR, USA). and a single-particle soot photometer (SP2, DMT, Longmont, CO, USA). Table 1 summarises the main instrument characteristics and the uncertainty estimates for reported $M_{BC}$. Ambient BC-containing particles were sampled by a Whole Air Inlet, suitable for long-term observations, and placed 2 m above the building rooftop of PDM measurement station. The air passing through the inlet was heated at $\sim 20°C$ to prevent the relative humidity from exceeding 20% (Nessler et al., 2003).

### 2.2.1   The Single Particle Soot Photometer (SP2)

The SP2 measures $M_{rBC}$ based on its incandescence capacity when heated to high temperatures. Its operating principle has been described previously (Baumgardner et al., 2012; Laborde et al., 2012b; Moteki and Kondo, 2010; Schwarz et al., 2006). To sum up, particles entering the instruments are passing through an Nd:YAG laser cavity, where they are heated up to 4000 K by the laser beam. If these particles contains rBC, they can then reach their incandescence point and produce a signal detected
by two Avalanche Photodiodes. Since species internally mixed with BC particles will evaporate before the rBC incandescence, the measured mass only takes into account the amount of rBC mass without interference from its potential coating. The intensity of this signal is proportional to the rBC mass. The incandescence peak height is converted to an individual rBC mass using





a calibration factor and then a rBC density of 1800 kg m$^{-3}$ (Bond and Bergstrom, 2006) is used to convert the rBC mass into an rBC mass equivalent diameter.

The calibration was performed using monodispersed fullerene soot (Alfa Aesar, lot #FS12S011) selected by a differential mobility analyzer.

The SP2 data were processed using the PySP2 code, a computer code written in Python that derives the rBC mass concentration and mixing state from SP2 measurements (Tinorua et al., 2023). The rBC mass was quantified from $\sim$ 0.68 to 320 fg, corresponding to $90 < D_{rBC} < 700$ nm. This size range was set by comparing the particle number concentration of the size-selected

fullerene soot particles measured by the SP2 against the one measured by a Condensation Particle Counter (CPC, model 3772, TSI Inc., Shoreview, USA). Typical rBC size distributions tend to fall a consistent larger range from a few nanometers to a few micrometers (Bond et al., 2013). To date, there have been three approaches to estimate and correct for the rBC mass undetected by the SP2. All these methods are based on fitting the measured rBC size distribution with lognormal distribution and extrapolating the measured rBC size distribution to larger and smaller sizes to estimate the "missing" rBC mass outside

the measurement range, hereafter referred as $R_{fit/meas}$ and calculated using Eq. 1:

$$R_{\text{fit/meas}} = \frac{M_{\text{rBC,fit}} - M_{\text{rBC,meas}}}{M_{\text{rBC,meas}}} \tag{1}$$

where $M_{rBC,fit}$ is the fitted rBC size distribution and $M_{rBC,meas}$ the measured rBC size distribution by the SP2.

(i) The most widely used approach is to fit the campaign average size distribution with a single and monomodal lognormal distribution and to derive a single correction factor of the mass concentration, hereafter called $R_{fit/meas}$ (Schwarz et al., 2006;

Laborde et al., 2012a; Metcalf et al., 2012; Zanatta et al., 2018; Liu et al., 2010; Ko et al., 2020). (ii) Another method consists in fitting the campaign average size distribution with a multimodal lognormal distribution using the sum of two to four modes (Cappa et al., 2019; Raatikainen et al., 2017; Zhao et al., 2019). (iii) A last procedure proposed in the present study consists in calculating a time-dependent correction factor by fitting with a sum of several lognormal modes the rBC size distribution averaged on a shorter time period than the campaign duration. This approach is supported by the fact that rBC size distribution

can vary as a function of the sources and aging processes (Cappa et al., 2019; Takahama et al., 2014). In this study, all three methods were applied in order to assess the sensitivity to the correction approach. Bounds of the fitting parameters $d_g$ and $\sigma_g$ were fixed as following : Mode 1 : $50 < d_g < 100$ nm and $1.2 < \sigma_g < 3$; Mode 2 : $150 < d_g < 250$ nm and $1.3 < \sigma_g < 2.9$; Mode 3 : $350 < d_g < 500$ nm and $1 < \sigma_g < 3$ with $d_g$ and $\sigma_g$ the geometric mean diameter and the geometric standard deviation, respectively. The resulting uncertainty on $M_{rBC}$ is estimated to be around 24.5 %, taking into account measurement

uncertainties on the $M_{rBC}$ correction (see section 3.1) and the relative precision of SP2-derived $M_{rBC}$ (unit-to-unit variability as determined by Laborde et al. (2012a)).

### 2.2.2 The thermo-optical analyzer Sunset

Weekly integrated filter-sampled particles were analyzed using an EC/OC analyser (Sunset Laboratory Inc., Tigard, OR, USA), hereafter referred to as Sunset. We used the EUSAAR–2 heating protocol with transmittance correction (Cavalli et al., 2010).





This protocol was specifically developed for aerosol typically encountered at European background sites and it was recently selected as the European standard thermal protocol to be applied in air quality networks for the measurements of total carbon (TC), organic aerosol (OC) and EC in particulate matter samples (European Committee for Standardisation Ambient air, 2017; EN16909:2017).

  The measurement principle is based on the different volatilisation temperatures of OC and EC (Bauer et al., 2009). Briefly,
aerosols were collected on a pre-burnt quartz fiber filter at PDM and thermally desorbed in the Sunset Analyzer following a temperature gradient. A first step allowed the OC desorption by progressively heating to 500-700°C in an inert atmosphere with pure helium (He). A second step brought the filter at higher temperature ($\sim 850$ °C) in an oxidizing atmosphere composed of 98% helium and 2% dioxygen to induce the EC desorption. At each temperature step the OC (in the inert atmosphere) and the EC (in the oxydised atmosphere) are oxydised to carbon dioxide and then catalytycally reduced to methane, which is quantified
by a nondispersive infrared (NDIR) detector, and associated to a mass of OC or $M_{EC}$.

  Due to temperature elevation, some OC can be pyrolised and thus be desorbed during the second step of the procedure, leading to an overestimation of the sampled $M_{EC}$. This artefact due to the so-called Pyrolytic Carbon (PyrC) is corrected using thermal-optical transmittance correction. The split-point, determined in order to separate EC and refractory OC, is defined as the time at which the transmission through the filter in the second step (oxidised atmosphere) during the EC and PC desorption equals
the transmission through the filter in the first step (inert atmosphere) before PC formation (i.e. initial value of transmission). The distinction between PyrC and native EC is based on two assumptions : (1) During the oxidative phase, PyrC is the first to completely evolve before native EC and (2) PyrC and native EC have the same mass absorption cross section (MAC) (Yang and Yu, 2002). Thus, the total OC mass on the filter is the sum of the four OC fractions (OC1, OC2, OC3 and OC4) and the total EC mass on the filter is the sum of the four EC fractions (EC1, EC2, EC3 and EC4) minus the PyrC mass determined
optically (which is converted in OC fraction).

  The Sunset calibration was performed using a sucrose ($C_6H_{12}O_6$) solution containing two different known TC surface loading (35.16 and 42.527 $\mu$gC cm$^{-2}$), spiked on blank filters. Several blank filters were analysed over the campaign and the resulting transmission intensity was introduced as an offset in the data. The limit of detection (LOD) of $M_{EC}$ is generally estimated using different methods based on filter blank measurements. The first one consists in averaging $M_{EC}$ across several blank
filter samples (Sciare et al., 2011). Bauer et al. (2009) estimated the LOD by calculating the 95[th] percentile off the standard deviation over zero air measurements. The method recommended by EN16909:2017 consists in calculating the average Sunset blank filter value from a high number of blank measurements and add two times its standard deviation (Jaffrezo et al., 2005; Karanasiou et al., 2020). The last method that was used here consists in taking three times the standard deviation of several blank measurements as proposed by Brown et al. (2019). By using 22 blank measurements, blank levels of 0.17 and 0.70 $\mu$gC
cm$^{-2}$ were obtained for EC and TC, respectively (using an average sampling volume of 440 m$^3$). The LOD on EC is similar to the one of 0.18 $\mu$gC. cm$^{-2}$ obtained by Zheng et al. (2014), who studied the variations of the LOD among different protocols, and close to the LOD of 0.1 $\mu$gC cm$^{-2}$ determined by Bauer et al. (2009).





### 2.2.3 The dual-spot aethalometer AE33

A dual-spot aethalometer (model AE33, Magee Scientific, USA) was used to quantify $M_{eBC}$. This instrument measures the
attenuation of light at seven wavelengths (370, 470, 520, 590, 660, 880 and 950 nm) through a filter where particles are
continuously collected. $M_{eBC}$ was calculated from the attenuation coefficient $\sigma_{ATN}$ measured at 880 nm because other light-
absorbing particles, such as brown carbon (BrC) and mineral dust, absorb significantly less at this wavelength (Samset et al.,
2018). In the aethalometer, the change in attenuation with time is caused by both the increasing mass of eBC deposited
on the filter (i.e. loading effect), the scattering by particles and filter matrix (Weingartner et al., 2003; Segura et al., 2014).
To overcome the loading effect, a compensation algorithm has been incorporated into the AE33 using an on-line dual-spot
technology (Drinovec et al., 2015).

Following Drinovec et al. (2015) $\sigma_{ATN}$ and $M_{eBC}$ can be derived by applying Eq. 2 and Eq. 3, respectively:

$$\sigma_{ATN} = \frac{S.\frac{\Delta ATN}{100}}{F(1-\zeta).(1-k\Delta ATN).\Delta T} \tag{2}$$

$$M_{eBC} = \frac{S.\frac{\Delta ATN}{100}}{F(1-\zeta).C.(1-k\Delta ATN).\Delta T.MAC} \tag{3}$$

where S is the filter surface area loaded with the sample, $\Delta$ATN the variation of attenuation on this surface on the spot, F
the measured flow rate passing through the instrument corrected by the leakage factor $\zeta$ (here equals to 0.01), C the multiple
scattering coefficient, k the loading factor parameter and $\Delta$t the sampling duration of aerosols in the aethalometer fixed at 20
minutes.

$M_{eBC}$ was first determined using the default instrumental filter constant C of 1.39 for the M8060 filter tape and MAC of 7.77
m² g$^{-1}$ at 880 nm. In Section 3.5, we have estimated the hourly and weekly C×MAC values that allow to better match the
$M_{eBC}$ with $M_{rBC}$ and $M_{EC}$ at PDM.

To limit biases caused by extreme values, $M_{eBC}$ under the lower detection limit of the AE33 (0.005 $\mu$g m$^{-3}$) and over the 95$^{th}$
percentile of the distribution were filtered before the analysis.

### 2.3 Other parameters

Aerosol scattering coefficients ($\sigma_{sca}$) were retrieved by an integrating nephelometer (model Aurora 3000, Ecotech Pty Ltd,
Knoxfield, Australia) at 450, 525 and 635 nm and the Scattering Angström Exponent (SAE) was then calculated between 450
and 635 nm ($SAE_{450-635}$). Absorption coefficients were derived from the attenuation measurements of the AE33 at 470 and
660 nm, and then converted to 450 and 635 nm using the Absorption Angström Exponent calculated between 370 and 470 nm
($AAE_{370-470}$) and between 590 and 660 nm ($AAE_{590-660}$), respectively.
200 To avoid biases linked to inlet artifacts and local pollution, periods where precipitation events, high humidity (95%) and high
CO concentrations (>200 ppb) occurred were removed from the dataset. Weekly averages were calculated from hourly SP2 and



AE33 data when at least 70% of the hourly data were available. All data were converted to standard temperature and pressure (STP) using local meteorological data.

## 3 Results

### 3.1 Sensitivity of rBC mass concentration to size distribution processing from the SP2 measurements

Figure 1 presents the campaign-average mass weighted rBC size distribution derived from the SP2. Two distinct peaks can be seen at $D_{rBC}$= 130 nm and $D_{rBC}$= 180 nm. A third coarse mode seems to raise around 490 nm. From this figure, it can clearly be seen that SP2 does not measure the rBC size distribution over its full size range. We evaluated three different approaches described in Section 2.2.1 to correct $M_{rBC}$ for the rBC mass undetected by the SP2. The extend to which the different approaches contributes to uncertainty on the overall $M_{rBC}$ was quantified by comparing the fitted to observed $M_{rBC}$ on the SP2 detection size range ($90 < D_{rBC} < 700$ nm). When using a 1-mode representation, the fitting procedure gave a mode centered on $D_g$ = 153 nm on average over the 2-year campaign while the three modes of the trimodal fit peaks at $D_{g,1}$ = 98 nm, $D_{g,2}$ = 177 nm and $D_{g,3}$ =377 nm. A first direct comparison between the fitting procedure shows that both approaches represent well the measurements for $D_{rBC}$ between 90 and 150 nm. However,the 1-mode fit overestimates $M_{rBC}$ for $D_{rBC}$ between 150 and 230 nm and completely misses the largest mode above 400 nm, while both are well represented by the trimodal approach. As a first conclusion, the trimodal curve generally better follows the measurements, and in particular for rBC diameter above 150 nm.

Figure 2 shows the ratio between $M_{rBC}$ estimated from the fitting approaches and derived from the observation ($M_{rBC,fit}/M_{rBC,meas}$) over the $D_{rBC}$ range covered by the SP2 for the different fitting approaches throughout the campaign (Figure 2a) and the overall statistical results (Figure 2b). Considering all data points from the campaign, the unimodal fit tends to slighly underestimate $M_{rBC}$ by around 1.6% regardless of the selected averaging time. When using a trimodal lognormal function, the overall statistics for these two quantities agree more closely within 0.4 %. However, both the variability and the systematic bias in the $M_{rBC,fit}/M_{rBC,meas}$ ratio are considerably smaller when fitting the rBC size distribution with a trimodal lognormal distribution and on a daily basis. The evolution of the $M_{rBC,fit}/M_{rBC,meas}$ ratio over the 2-year measurement period shown in Fig. 2a illustrates that $M_{rBC,fit}$ differs regularly from $M_{rBC,meas}$ by more than 5% for every fitting procedure, and even more on autumn and winter. In fact, as shown in Figure S1 in the Supplements, the averaged rBC size distribution observed in autumn and winter are more noisy than the one in summer and spring. In particular, the second mode usually peaking at around 170-180 nm is less clearly defined in autumn and winter with some outliers above the general distribution. The third coarse mode at $D_{rBC} \sim 500$ nm exhibits also a higher variability in cold seasons compared to the ones fitted in spring and summer. Tinorua et al. (2023) showed that winter was characterised by lower $M_{rBC}$ with some rBC-containing particles injected from the boundary layer during the day, compared to summer when an additional long-range transport of rBC-containing particles occurred. Thus, the noisier rBC size distributions in winter and autumn could be due to the low daily $M_{rBC}$ which leads to greater uncertainties in the fitting procedure, and/or more sporadic variability in $M_{rBC}$ due to the PBL dynamics in winter. Overall these results suggests that neglecting the day-to-day variability of rBC size distribution at PDM may lead to an overestimation of $M_{rBC}$.



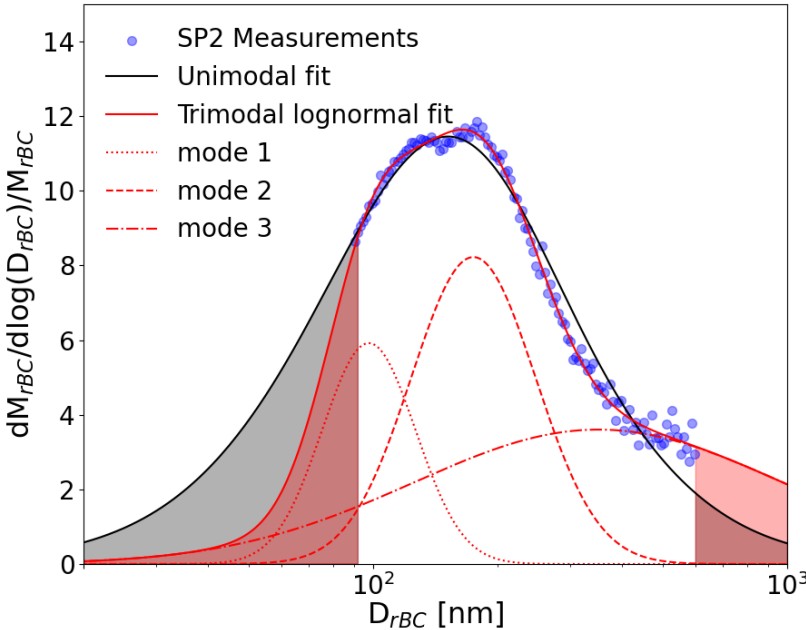

**Figure 1.** rBC core size distribution averaged over the campaign and fitted with a unique (in black lines) or a sum of three log-normal functions (in red line). The three modes are in red dotted lines. Blue dots represent the SP2 measurements over its size detection range. The black and red shaded areas represent the rBC mass fraction missed by the SP2 and recalculated when applying a unimodal or a trimodal fitting procedure, respectively. Data were normalised by the sum of $M_{\mathrm{rBC}}$ and averaged over the campaign.

Although the overall bias between $M_{\mathrm{rBC,meas}}$ and $M_{\mathrm{rBC,fit}}$ remained low ( < 2% on average over the 2-year campaign) re-
gardless the approach chosen, larger differences in rBC size distribution can be observed for $D_{\mathrm{rBC}}$ < 90 nm (Fig. 2a). The
extrapolation of rBC size distribution towards lower and larger sizes lead to a missed mass fraction $R_{\mathrm{fit/meas}}$ calculated using
Eq. 1 of around 25% and 15% for the unimodal and multimodal fit, respectively. Therefore, in the following, all reported $M_{\mathrm{rBC}}$
are corrected with the multimodal fit on a daily basis. It can be seen in Figure 1 that the largest differences between the 1-mode
and the 3-modes fitting approaches occurs for $D_{\mathrm{rBC}}$ < 90 nm. It is important to note that the lack of measurements below 90 nm
complicates the estimation of the missing mass fraction. Since it is possible that discrepancies between instruments arise from
the extrapolation of rBC size distribution towards lower and larger sizes, the influence of the estimated fraction of undetected
rBC particles on the biases between instruments will be investigated in Section 3.4.




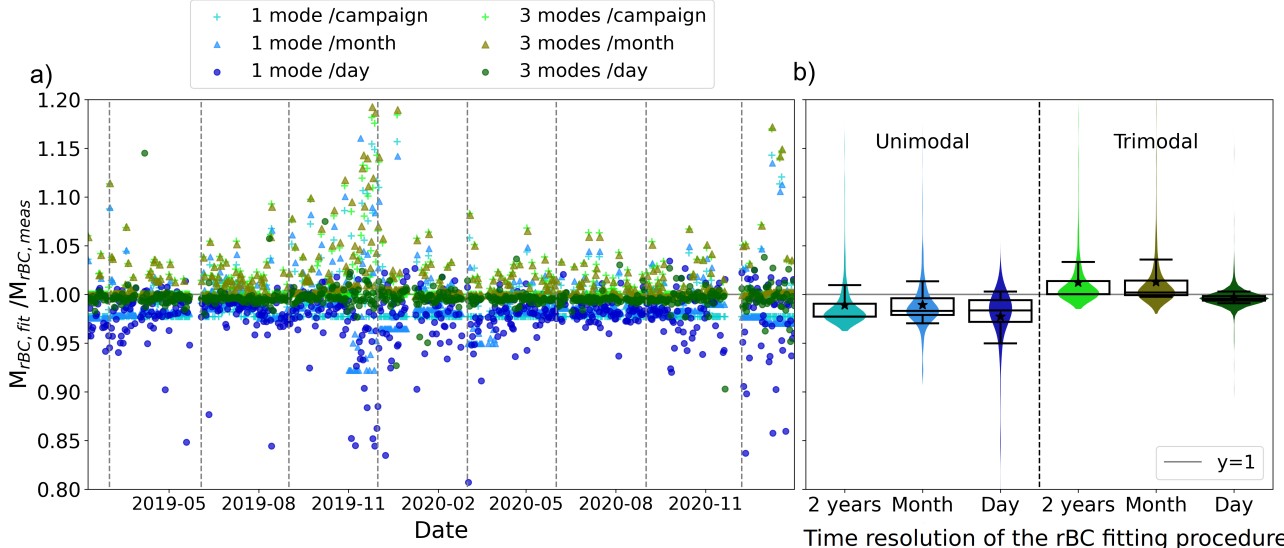

**Figure 2.** a) Time series and b) statistics of $M_{rBC,fit}/M_{rBC,meas}$ calculated between 90 and 700 nm when representing the rBC size distribution with one (lightblue and darkblue) or three (light green and green) modes and when fitting on the mean campaign (crosses), monthly (triangles) or daily (plain circles) size distribution. Violin plot represent the probability density function of $M_{rBC,fit}/M_{rBC,meas}$. Boxes and whiskers represent the $25^{th}$, $75^{th}$, $10^{th}$ and $90^{th}$ percentiles. Lines and stars shows medians and mean values.

## 3.2 Overview of rBC, EC and eBC mass concentrations measured by the SP2, Sunset and AE33 and their relationship

Figure 3 shows the temporal variation and frequency distribution of $M_{rBC}$, $M_{EC}$ and $M_{eBC}$, which are respectively BC mass concentrations measured by the SP2, Sunset and AE33 from January 2019 to February 2021. Table S1 in the Supplements reports statistical analysis of $M_{rBC}$, $M_{EC}$ and $M_{eBC}$. Over the whole period the mean values (± GSD) of $M_{EC}$, $M_{rBC}$, and $M_{eBC}$ were 54.7 (± 25.3), 36.4 (± 28.4), and 75.5 (± 54.3) ng m$^{-3}$, respectively. All techniques respond to seasonal variations of the atmospheric changes in BC levels. Average concentrations were around 2-4 times (depending on the measurement technique) higher in summertime than in wintertime, with monthly averages ranging from a minimum of 14.2-44.8 ng m$^{-3}$ in December to a maximum of 65.6-142.3 ng m$^{-3}$ in July. This seasonal variation is similar to those reported at the high-altitude European sites of Jungfraujoch in Switzerland for $M_{eBC}$ measurements (Bukowiecki et al., 2021) and Montseny in Spain for $M_{EC}$ measurements (Zanatta et al., 2016). At the PDM, this seasonality was attributed to the combined effects of less precipitation and a larger contribution of long-range transport from biomass burning sources during summer (Tinorua et al., 2023).

Figure 4 shows scatterplots of time-resolved relationship between $M_{rBC}$, $M_{EC}$ and $M_{eBC}$ over the campaign using distinct colors for each month. Positive correlations were observed between all instruments with Pearson's r values ranging from 0.66 to 0.80. All linear regressions were based on an assumption of a zero intercept.





Considering all data points, the largest bias is observed between the AE33 and the SP2 measurements (fig. 4c.), with $M_{eBC}$

higher by a factor 1.96 than $M_{rBC}$ on average. The good Pearson's r value of 0.80 shows that the bias is systematic. The correlation slope is almost parallel with the 1:1 line, meaning that there is an offset error of $M_{eBC}$ compared to $M_{rBC}$. Considering the colors of the points, there is no link between the time of the year and the value of the bias.

The SP2 and Sunset measurements appear to have the best agreement with a correlation slope of 1.17 between $M_{EC}$ and $M_{rBC}$. As shown in Figure 4a, the correlation between the Sunset and the SP2 measurements degrades for $M_{EC}$ lower than 50

ng m$^{-3}$. The threshold $M_{EC}$ value can also be observed to a lesser extent on the relationship between the AE33 and Sunset measurements with a slight break in the correlation slope below 50 ng m$^{-3}$ (Figure 4b). These results point to a loss of Sunset sensitivity below this limit value. The reasons behind these results will be investigated in the next section.





**Figure 3.** Time series (left) and statistical analysis (right) of (a) $M_{EC}$, (b) $M_{rBC}$ and (c) $M_{eBC}$ measured during the campaign. Dots connected with lines represent monthly averages and little dots represent hourly data. For (b) and (c), Probability Distribution Functions of $M_{rBC}$ and $M_{eBC}$ was calculated over hourly data in colors and over resampled data on the time resolution of the Sunset in black lines. (d) Seasonal boxplot of $M_{EC}$, $M_{rBC}$ and $M_{eBC}$.





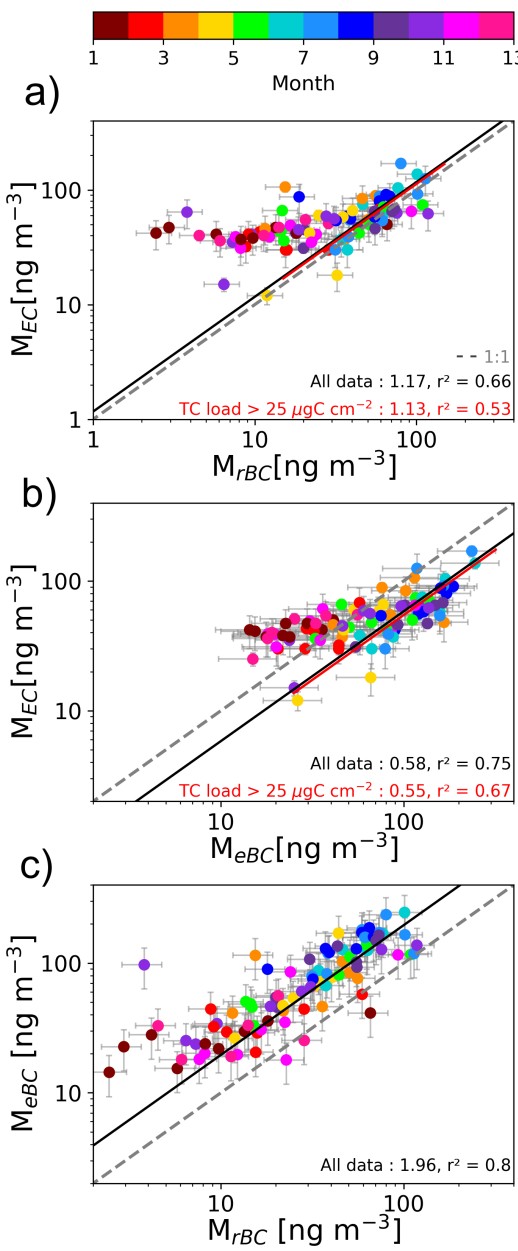

**Figure 4.** Scatterplot of the relationship between (a) $M_{EC}$ versus $M_{rBC}$, (b) $M_{EC}$ versus $M_{eBC}$ and (c) $M_{eBC}$ versus $M_{rBC}$ over to the two measurement period. The correlation coefficients are shown for each plot. *For $M_{EC}$ vs. $M_{rBC}$ and $M_{EC}$ vs. $M_{eBC}$, the slope excluding data where the TC load was lower than 25 $\mu$gC cm$^{-2}$ has been calculated.



### 3.3 Biases on EC mass concentration measured by the Sunset : filter underloading and charring effect

The main challenges in isolating EC from TC analysis are the possible artifacts during OC/EC separation. In a study from the
EMEP Co-operative Programm for Monitoring and Evaluation of the Long-range Transmission of Air Pollutants in Europe im-
plying 36 instruments measuring $M_{EC}$ over Europe (EMEP/CCC – Report 1/2018, https://projects.nilu.no/ccc/reports/cccr1_2018_
Data_Report_2016_FINAL.pdf), bias is mentioned in the determination of the OC/EC split point at low TC concentrations,
which usually led to an overestimation of $M_{EC}$. However there is no detail provided to explain the effects.

Figure 5a shows how the filter TC loading influence the biases between $M_{EC}$ and $M_{rBC}$. The same analysis has not been
carried out for $M_{EC}$ to $M_{eBC}$ ratio because of the multiple sources of biases in $M_{eBC}$ (i.e. section 3.5). A systematic positive
bias and a wide dispersion of the $M_{EC}/M_{rBC}$ ratio can be observed at TC contents below 25 $\mu$gC cm$^{-2}$ (Mean ± GSD of
3.08 ± 2.71). Above this TC value, no significant dependence on the filter loading can be distinguished with a lower mean (±
GSD) $M_{EC}/M_{rBC}$ ratio of 2.35 ± 3.29. A large fraction of samples (42%) with TC loading below 25 $\mu$gC cm$^{-2}$ was measured
in winter. When data with a TC load below 25$\mu$gC cm$^{-2}$ are eliminated, the bias between $M_{rBC}$ and $M_{EC}$ is reduced, with
a decrease in the value of the slope from 1.17 to 1.13 (see Fig. 4). By contrast, excluding data with a TC load below 25$\mu$gC
cm$^{-2}$ leads to a slope of $M_{EC}$ vs. $M_{rBC}$ farthest from unity (0.55 against 0.58 considering all data, see Fig. 4). This is for sure
due to the compensating effect of data with a TC load lower than 25$\mu$gC cm$^{-2}$ on the overestimation of $M_{eBC}$ compared to
$M_{EC}$, thus leading to a slope slightly closer to 1.

Our results are consistent with the sharp reduction in the repeatability and reproducibility at low TC loadings (below 10
$\mu$gC cm$^{-2}$) reported during an inter-laboratory comparison for the measurement of $M_{EC}$ performed within the European
project ACTRIS-2 on ambient aerosol samples collected at a regional background site in Italy (EMEP/CCC–Report 1/2018).
Conversely, Pileci et al. (2021) did not find any increased random noise or systematic bias caused by low TC surface loading.
However, Fig S2 in the Supplement shows that the superimposition of our data with theirs indicate a similar trend with a wide
dispersion of $M_{EC}/M_{rBC}$ ratio below 25 $\mu$gC cm$^{-2}$. As explained in Zheng et al. (2014), the threshold TC load for accurate
thermo-optical analysis can vary with location and season due to the variation of thermal properties among carbonaceous
particles collected on the filter.

We further investigate the potential causes of the Sunset bias with a special focus on the charring correction used to derive
$M_{EC}$ with the EUSAAR-2 protocol. Optical correction is an essential component of the thermo-optical method to remove
measurement artifacts in OC and EC caused by charring of some OC components. Without correction, the charred fraction of
OC, also called PyrC, would be reported as part of EC, leading to an overestimate of $M_{EC}$. Charring depends on many factors,
including the amount and type of organic compounds, temperature steps in the analysis, the residence time at each temperature
step, and the presence of certain inorganic constituents or BrC (Yu et al., 2002; Subramanian et al., 2007; McMeeking et al.,
2009).

Figure 5b presents the $M_{EC}/M_{rBC}$ ratio as a function of the $M_{EC}/M_{TC}$ ratio to investigate if biases arise from the split between
OC and EC. No dependence of $M_{EC}/M_{rBC}$ to $M_{EC}/M_{TC}$ ratio can be observed. This result differs from that obtained by Pileci
et al. (2021) for five different field campaigns performed with different instruments. This could suggest that the quantification





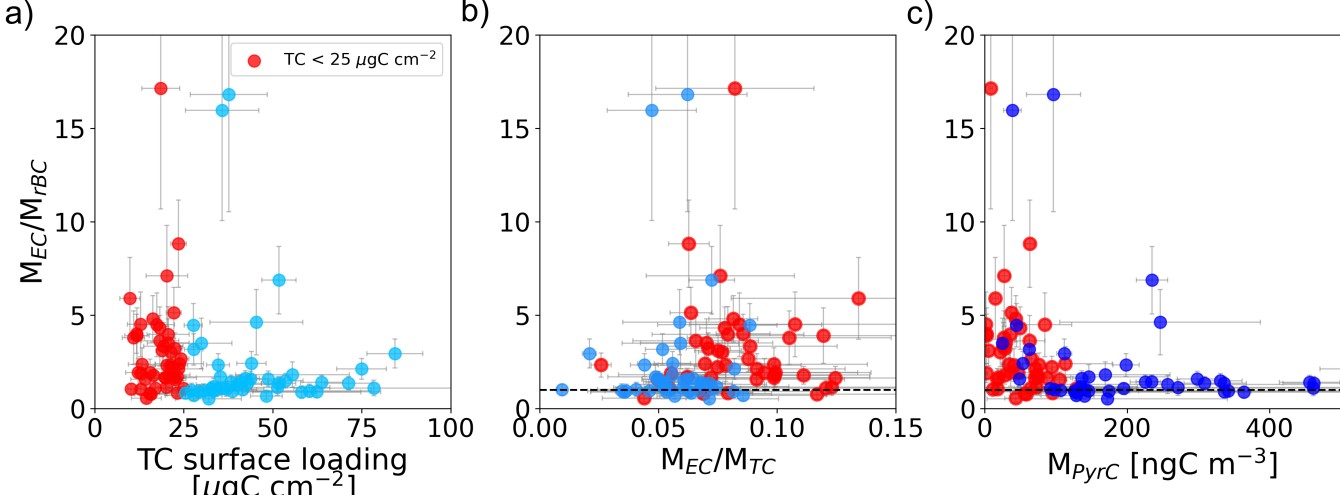

**Figure 5.** $M_{EC}/M_{rBC}$ as a function of a) Total Carbon surface loading, b) $M_{EC}/M_{TC}$ and c) $M_{PyrC}$. Points have a weekly time resolution. Red circles highlights data where the TC loading is under $25 \mu g\ cm^{-2}$

of $M_{EC}/M_{TC}$ ratio is more controlled by the instrument characteristic and set-up than the instrument-specific analysis-by-analysis variability.

Two distinct different patterns can be distinguished for $M_{EC}/M_{rBC}$ as a function of $M_{PyrC}$ in Figure 5c. A wide dispersion of

$M_{EC}/M_{rBC}$ ratio for $M_{PyrC}$ below 140 ngC cm$^{-3}$ can be observed with a mean ($\pm$ GSD) of 3.17 ($\pm$ 3.40), whereas $M_{EC}/M_{rBC}$ are closer to 1 (1.63 $\pm$ 1.29) above this threshold value. All the data with $M_{PyrC}$ above 140 ngC cm$^{-3}$ exhibited a TC loading over 25 $\mu g C\ cm^{-2}$. By contrast below $M_{PyrC}$ of 140 ngC m$^{-3}$ most samples (73%) exhibit a TC loading lower than 25 $\mu g C$ cm$^{-2}$. Interestingly the remaining samples with $M_{PyrC}$ below 140 ngC cm$^{-3}$ and TC loading above 25 $\mu g C\ cm^{-2}$ show a mean $M_{EC}/M_{rBC}$ value of around 3.40. This result indicates a possible underestimation of $M_{PyrC}$ for these samples that could

explain some of the bias between $M_{EC}$ and $M_{rBC}$.

Another possible measurement artifact can arise from the presence of dust and BrC particles as mentioned previously by Liu et al. (2022) and Karanasiou et al. (2020). However, as will be explained in section 3.5, the sporadic nature of dust events at PDM compared to the long sampling duration of the Sunset did not allow to identify samples with significant Saharan dust contribution. Furthermore, Tinorua et al. (2023) found a very low contribution of BrC to the aerosol absorption at PDM.

Ammerlaan et al. (2015) also highlighted a possible influence of the laser stability on $M_{EC}$ bias. Here, a deep analysis of the blank filters did not reveal laser instabilities. In addition, the analysis of the baseline of the transmission signal in the thermograms did not allow us to identify potential causes of the overestimation of $M_{EC}$.

In the following, all Sunset data for which TC content is lower than 25 $\mu g C\ cm^{-2}$ have been eliminated of the analysis.





### 3.4 Biases on rBC mass concentration measured by the SP2 due to the presence of undetected small/large
rBC-containing particles

As shown in section 3.1 and Figure 1, the limited SP2 detection range ($90 < D_{rBC} < 700$ nm) can lead to an underestimation of $M_{rBC}$. In this section we investigate if the $M_{EC}/M_{rBC}$ mass discrepancy could be partially explained by the presence of BC particles outside the SP2 detection size range. $M_{rBC}$ has been corrected for the missing mass concentration outside the SP2 detection range using an extrapolation method based on daily multi-modal fits to the measurement (i.e. section 2.2.1 and 3.1).

Figure 6 shows the campaign-averaged mass-weighted rBC size distribution classified into four ranges of fractional amount of missing mass fraction ($R_{fit/meas}$). $R_{fit/meas}$ increases strongly as the proportion of mode 1 (i.e. centered at around 100 nm) increases. This mode 1 becomes predominant when $R_{fit/meas}$ exceed 0.1 while the mode 2 (i.e. centered at around 180 nm) becomes secondary. Meanwhile, the proportion of the mode 3 (i.e. centered at around 500 nm) remain rather constant for all ranges of $R_{fit/meas}$. As shown in Table 2, the extrapolated mass fraction under the lower detection range of the SP2 ($D_{rBC}$

$< 90$ nm) increases from 7.9 to 19 % as $R_{fit/meas}$ increases from values $\leq 0.1$ to values $> 0.3$. By contrast, above the higher detection range ($D_{rBC} > 700$ nm), the extrapolated mass fraction is very low with values around 2.5 % and is not correlated to $R_{fit/meas}$ values. Overall, these results show that the presence of small rBC particles below the lower detection limit of the SP2 is the main contributor to the extrapolation calculations, whereas the contribution of large rBC particles above the higher detection limit is rather negligible.

Given the remote location in the Pyrenees and apparent distance from fresh BC source regions of the PDM site, it is expected that rBC particles sampled at this site are aged with relatively large sizes. Regarding the large contribution of ultrafine rBC particules, their presence at PDM is surprising, but could be explained by two hypotheses. First, it is possible that some periods with local influences of rBC emission may still remain despite the filtering of isolating spikes of CO (i.e. Section 2). However, the $R_{fit/meas}$ shows no correlation with $M_{rBC}$ (see figure S3 in the Supplements), meaning that the presence of ultrafine rBC

does not preferentially occur during local pollution event when $M_{rBC}$ were sporadically very high. In fact, an opposite trend can be observed with higher $R_{fit/meas}$ ($> 0.2$) under low $M_{rBC}$ conditions ($< 30$ ng m$^{-3}$). Second, these ultrafine rBC-containing particles could be produced by aviation emissions. Modal diameters of nonvolatile particle size distributions in aircraft turbine exhaust range from 15 to 40 nm (Lobo et al., 2015; Durdina et al., 2017, 2019) while unfiltered gasoline direct injection and diesel engines have larger count mean diameter values ranging from 50 to 100 nm (Burtscher, 2005; Momenimovahed and

Olfert, 2015). Recently, BC mass emissions was estimated to be around 100-1000 g/km² per year above the Pyrenees region (Zhang et al., 2019b) if only taking into account the global civil aviation. Tinorua et al. (2023) showed that the dominant air mass origin at PDM is the North-Atlantic Ocean, where around 14% of the total BC mass emissions from civil aviation occurs (Zhang et al., 2019a).





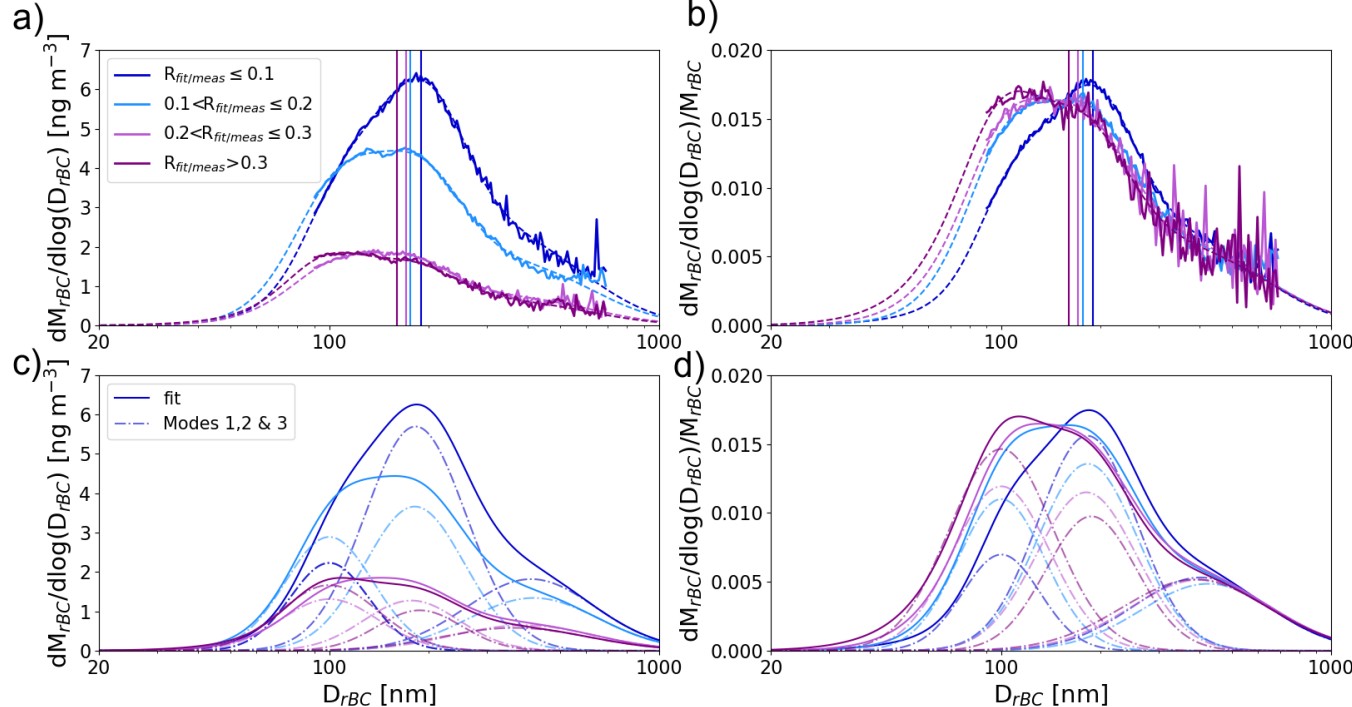

**Figure 6.** a) Mass size distribution of rBC core measured by the SP2 colored as a function of the missing mass correction factor $R_{\mathrm{fit/meas}}$. Vertical lines show the geometric diameter of each rBC mass size distribution. The fitting procedure is represented by dotted lines. b) same as a) but normalised by the total rBC mass. c) Same as a) but showing the position of the three modes. d) Same as c) but normalised by the total rBC mass.

In order to estimate the extent to which the extrapolation in rBC size distribution contributes to uncertainty in the overall
$M_{\mathrm{rBC}}$, Figure 7 shows the mass-weighted rBC size distribution classified into different ranges of $M_{\mathrm{EC}}/M_{\mathrm{rBC}}$ ratio : a significant negative bias ($M_{\mathrm{EC}}/M_{\mathrm{rBC}} \leq 1 - \Delta(M_{\mathrm{EC}}/M_{\mathrm{rBC}})$, with $\Delta(M_{\mathrm{EC}}/M_{\mathrm{rBC}})$ representing the uncertainty in $M_{\mathrm{EC}}/M_{\mathrm{rBC}}$), an agreement within the uncertainty range ( $1 - \Delta(M_{\mathrm{EC}}/M_{\mathrm{rBC}}) < M_{\mathrm{EC}}/M_{\mathrm{rBC}} \leq 1 + \Delta(M_{\mathrm{EC}}/M_{\mathrm{rBC}})$) and a significant positive bias ($M_{\mathrm{EC}}/M_{\mathrm{rBC}} > 1 + \Delta(M_{\mathrm{EC}}/M_{\mathrm{rBC}})$). An examination of the shape of mass size distribution of rBC core in figure 7 did not reveal any variability for the different ranges of $M_{\mathrm{EC}}/M_{\mathrm{rBC}}$ values. From these results we can not conclude any influence of
the SP2 limiting size range on the discrepancy between the SP2 and Sunset. However, it is important to note that the modal diameters and widths for Mode 1 and 3 are particularly uncertain as these modes occur near the lower and higher detection limit of the SP2. Uncertainties in the position and width for these mode may contribute to uncertainty in the total $M_{\mathrm{rBC}}$. In addition it is possible that (1) the extrapolation of the first mode peaking at $\sim$ 100 nm is inaccurate for masses lower than 90 nm as a large part of this mode occurs below the lower size detection limit of the SP2 or (2) the SP2 missed the detection of a
mode that is centered at lower diameter than the lower limit of detection of the SP2.





| $R_{\mathrm{fit/meas}}$ | Total fitted mass $M_{\mathrm{rBC}}$ [ng m$^{-3}$] | $M_{\mathrm{rBC}}$ for $D_{\mathrm{rBC}}$ <90 nm [ng m$^{-3}$] | Extrapolated %age to $D_{\mathrm{rBC}}$ <90 nm | $M_{\mathrm{rBC}}$ for $D_{\mathrm{rBC}}$ >700 nm [ng m$^{-3}$] | Extrapolated %age to $D_{\mathrm{rBC}}$ >700 nm |
|---|---|---|---|---|---|
| ≤ 0.1 Mean (GSD) | 3.74 (4.16) | 0.295 (0.366) | 7.90 (8.79) | $8.49 \cdot 10^{-5}$ ($9.75 \cdot 10^{-5}$) | 2.27 (2.35) |
| ]0.1 ;0.2] Mean (GSD) | 3.00 (2.62) | 0.385 (0.354) | 12.8 (13.5) | $6.79 \cdot 10^{-5}$ ($6.56 \cdot 10^{-5}$) | 2.26 (2.51) |
| ]0.2 ;0.3] Mean (GSD) | 1.29 (1.54) | 0.195 (0.234) | 15.2 (15.2) | $3.10 \cdot 10^{-5}$ ($4.27 \cdot 10^{-5}$) | 2.41 (2.76) |
| >0.3 Mean (GSD) | 1.28 (1.91) | 0.247 (0.447) | 19.4 (23.5) | $2.52 \cdot 10^{-5}$ ($3.57 \cdot 10^{-5}$) | 1.97 (1.87) |
| All Mean (GSD) | 2.46 | 0.290 (0.360) | 11.8 (12.1) | $5.49 \cdot 10^{-5}$ ($6.97 \cdot 10^{-5}$) | 2.23 (2.34) |

**Table 2.** Extrapolated mass fraction of $M_{\mathrm{rBC}}$ outside the SP2 size detection range for each $R_{\mathrm{fit/meas}}$.

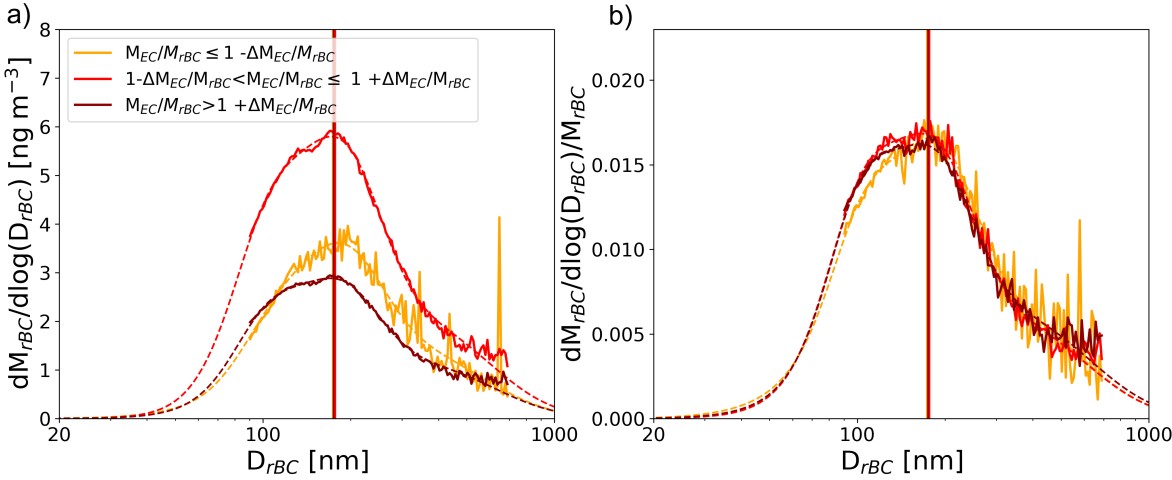

**Figure 7.** a) Mass size distribution of rBC core measured by the SP2, colored by $M_{\mathrm{EC}}/M_{\mathrm{rBC}}$, grouped by ranges of values. b) is the same as a) but normalised by the total rBC mass. Vertical lines highlights geometrical diameter corresponding to the color of the $M_{\mathrm{EC}}/M_{\mathrm{rBC}}$ range.

## 3.5 Biases on eBC mass concentration measured by the AE33

The main uncertainties in the $M_{\mathrm{eBC}}$ inferred from the AE33 measurement are the MAC used to calculate $M_{\mathrm{eBC}}$ from the absorption coefficient and the correction due to multiple scattering of particles sampled on the filter (Arnott et al., 2005; Bond et al., 1999; Collaud Coen et al., 2010; Weingartner et al., 2003). The multiple-scattering correction factor C depends on

the optical properties of the aerosol collected on the filter and the filter tape used. A constant MAC value of 7.77 m² g$^{-1}$ is





recommended by the AE33's manufacturer to convert $\sigma_{abs,880}$ to $M_{eBC}$, which is representative of optical properties of fresh BC particles (Bond et al., 2013). Nonetheless a wide range of MAC of BC from 5.9 to 54.8 m² g$^{-1}$ at 880 nm has been reported from field and laboratory measurements (Wei et al., 2020). This variability is due to the diversity of BC microphysical and chemical properties, which are related to their emission sources (Schwarz et al., 2008) and the effects of ageing processes
during the transport in the atmosphere (Ko et al., 2020; Sedlacek et al., 2022; Peng et al., 2016).

We first recalculated hourly (weekly) C×MAC values obtained at PDM by dividing $\sigma_{ATN}$ at 880 nm by $M_{rBC}$ ($M_{EC}$). Figure 8a shows that daily C×MAC values calculated with $M_{rBC}$ are around 2 to 3 times higher than those recommended by the manufacturer, with a median value of 27.8 m² g$^{-1}$, against 10.8 m² g$^{-1}$ for the constructor's value. In addition, a clear seasonal pattern can be observed with median values of 24.5 m² g$^{-1}$ and 31.3 m² g$^{-1}$ in spring and summer, respectively. C×MAC
values calculated using $M_{rBC}$ from the SP2 are around 21 % higher than those obtained using $M_{EC}$ from the Sunset (Figure 8c) with median values of 25.2 and 20.8 g$^{-1}$, respectively, despite averaging SP2 over the same time resolution as Sunset (Figure 8b). However, similar seasonal variability in the C×MAC values was obtained using $M_{EC}$ and $M_{rBC}$, although no statistical values could be obtained in winter due to too low $M_{EC}$ values during this season. This is consistent with the seasonal trend of C values obtained at Montsec d'Ares in the Spanish Pyrenees by Yus-Díez et al. (2021) using $\sigma_{abs}$ measured by a
MAAP (Multi-Angle Absorption Photometer). In addition, Pandolfi et al. (2014) found also higher MAC values at 637 nm in summer than in spring at this site.

In order to investigate the cause of the seasonal variation of C×MAC, we plotted on Figure 9 the correlation between C×MAC and $\Delta M_{rBC}/\Delta CO$ ratio for each season. The $\Delta M_{rBC}/\Delta CO$ ratio has been shown to be a good tracer of the rBC combustion source and wet deposition (Baumgardner et al., 2002; Taylor et al., 2014). Here $\Delta M_{rBC}$ and $\Delta CO$ were estimated using the
approach presented in Tinorua et al. (2023). Briefly, the hourly $\Delta CO$ were obtained by subtracting the hourly CO concentrations by the background CO concentrations estimated from the rolling 5[th] percentile of the values on a 14-day time window. $\Delta M_{rBC}$ was considered to be equal to $M_{rBC}$, thus assuming that the background $M_{rBC}$ is zero. Air masses for which precipitation occurred along 72-h back trajectories performed with the Hysplit model were removed in order to investigate the influence of rBC sources only. Lower C and MAC values are generally observed in the literature for rBC-containing particles emitted
from fossil combustion compared to those emitted from biomass combustion (Laing et al., 2020; Sedlacek et al., 2022; Healy et al., 2015; McMeeking et al., 2014; Denjean et al., 2020). A significant increase of C×MAC can be observed in Fig. 9 for $\Delta M_{rBC}/\Delta CO < 2$ ng m$^{-3}$ ppbv$^{-1}$ in every season except for spring, suggesting that a BC source-dependant correction should be applied to the AE33.





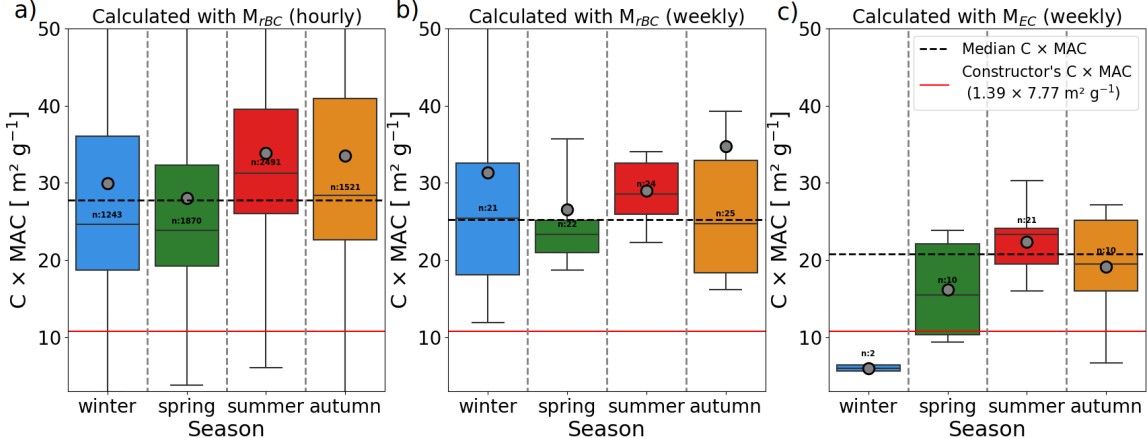

**Figure 8.** C×MAC as a function of the season calculated with a) $M_{rBC}$ in an hourly basis, b) $M_{rBC}$ averaged over the sampling period of the Sunset, and c) $M_{EC}$, excluding TC loading < 25 $\mu$gC cm$^{-2}$. Constructor's values are represented with a red solid line while median values are represented with a black dashed line. For b), the median is calculated excluding winter's value to be in agreement with c). Boxes, lines, black dots and whiskers indicate 25$^{th}$ percentile, 75$^{th}$ percentile, median, mean, 10$^{th}$ percentile and 90$^{th}$ percentile, respectively.

The different trend observed between C×MAC and $\Delta M_{rBC}/\Delta CO$ in spring may be due to a measurement artifact during

this season. In order to study the influence of other co-existing light-absorbing particles on C×MAC, a classification of the dominant aerosol type sampled at the PDM was performed. This classification, detailed in Tinorua et al. (2023), is based on the daily analysis of the spectral aerosol optical parameters AAE and SAE. Aerosols with AAE > 2 and SAE < 0.25 were classified as dust-dominated, aerosols with AAE > 2 and SAE > 1.5 were classified as BrC and AAE > 1.5 characterised aerosol mixtures containing dust particles and/or BrC (Kirchstetter et al., 2004; Lack and Cappa, 2010).

Figure 10b shows $M_{eBC}/M_{rBC}$ ratio as a function of the season and the dominant aerosol type. The same analysis could not be carried out for $M_{eBC}/M_{EC}$ ratio due to the short duration of the dust events reaching PDM (<1-2 days) compared to the duration of aerosol sampling for Sunset analyses (1 week). The level of agreement between $M_{eBC}$ and $M_{rBC}$ over the 2-year campaign degrades by a factor of 2 when aerosols were dominated by dust particles (averaged $M_{eBC}/M_{rBC}$ ± STD of 6.7 ± 3.6 and 3.2 ± 6.3 during and outside dust events, respectively). The bias is the greatest in spring with $M_{eBC}/M_{rBC}$ ratio

reaching 8.6 ± 3.7 (see Fig 10a). This seasonality is due to a stronger influence of dust events transported in the Pyrenees in spring (Tinorua et al., 2023). It should be noted that no BrC-dominated events were observed at PDM, probably due to their low lifetime in the atmosphere (around 1 day) (Forrister et al., 2015; Wong et al., 2019). In addition, no increase of $M_{eBC}/M_{rBC}$ ratio is observed for aerosols composed of a mixing of rBC with dust and/or BrC particles, suggesting that only the predominance of dust particles in the aerosol lead to significant biases in $M_{eBC}$ retrieval. Previous studies showed that a higher C

value should be applied for dust samples (Yus-Díez et al., 2021; Di Biagio et al., 2017). Using the same instrumentation as this study, Yus-Díez et al. (2021) showed that a C value of around 3.95 should be used for correcting multiple scattering artefacts




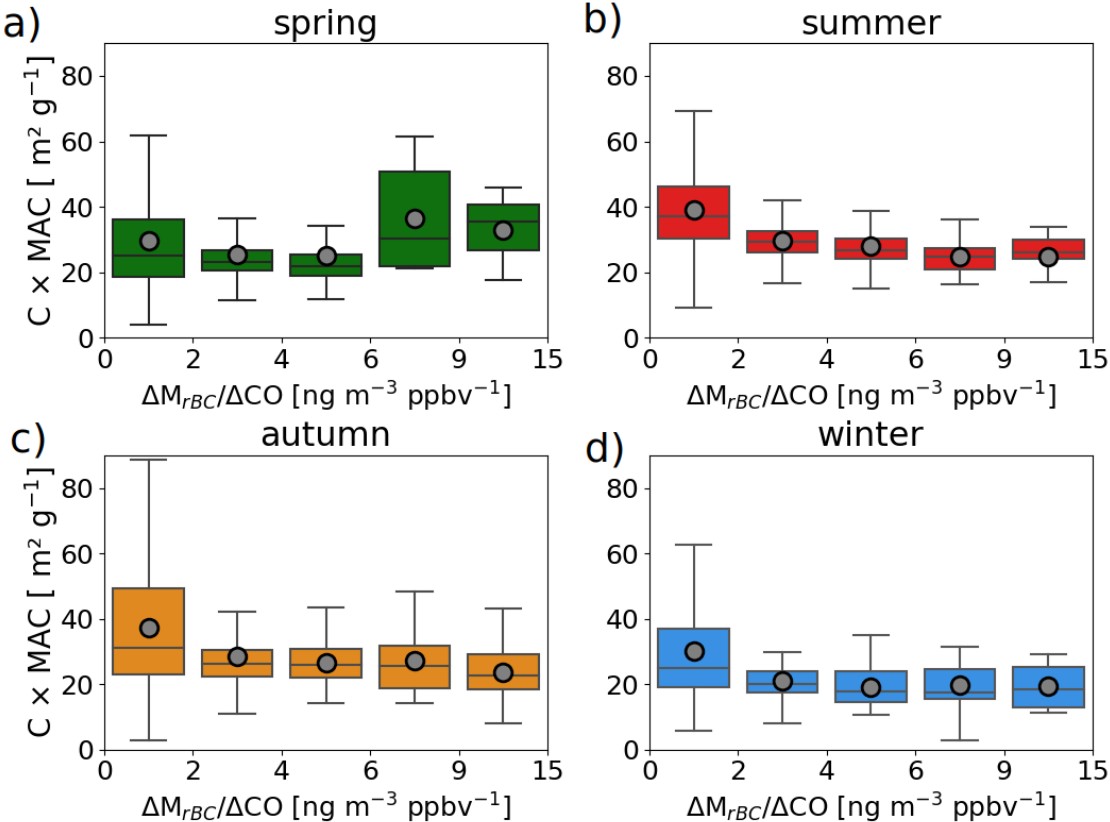

**Figure 9.** a) to d) C×MAC as a function of the $\Delta M_{rBC}/\Delta CO$ emission ratio for spring, summer, autumn and winter, respectively. Boxes, lines, black dots and whiskers indicate 25[th] percentile, 75[th] percentile, median, mean, 10[th] percentile and 90[th] percentile, respectively.

in the AE33 during Saharan dust outbreaks. As shown in the equation (2), an increase of C value would lead to decrease of $M_{eBC}$ values and thus a decrease of $M_{eBC}/M_{rBC}$ ratio from 6.7 to 2.3. Thus, a C readjustment taking into account the presence of dusts significantly improves the $M_{eBC}/M_{rBC}$ ratio. Nonetheless, a bias is still present even without the presence of dusts, suggesting an inappropriate MAC value regarding eBC measured at PDM.






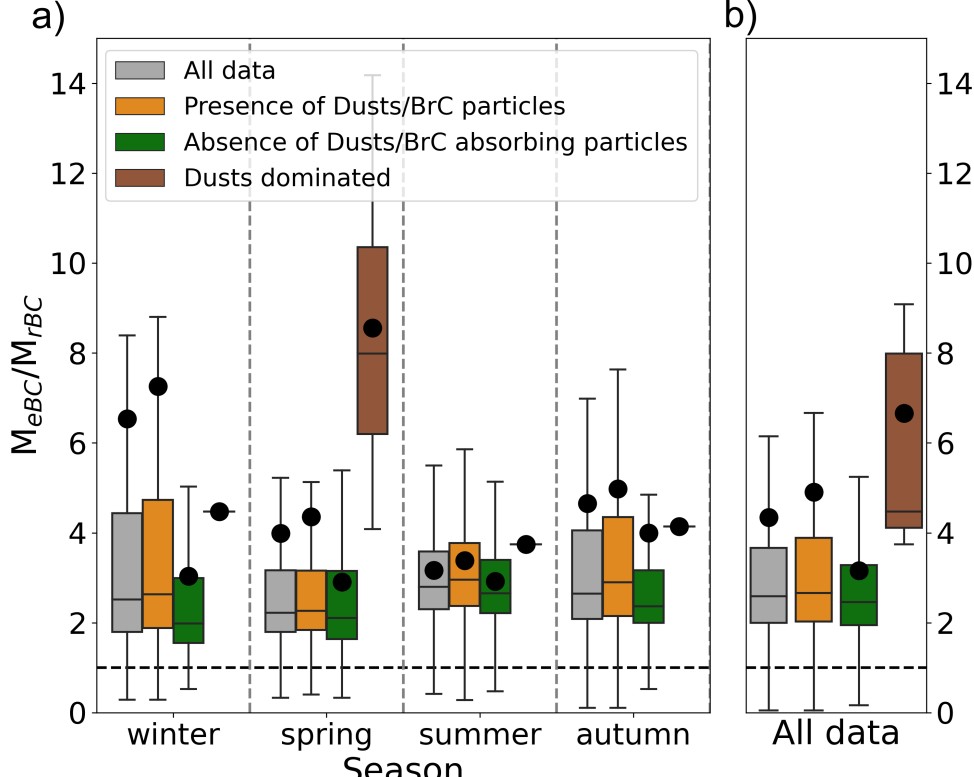

**Figure 10.** $M_{eBC}/M_{rBC}$ as a function of the dominant aerosol type a) as a unction of the season and b) over the 2-year measurement campaign. Boxes, lines, black dots and whiskers indicate $25^{th}$ percentile, $75^{th}$ percentile, median, mean, $10^{th}$ percentile and $90^{th}$ percentile, respectively.

## 4 Summary, conclusions and recommendations

Three of the most widely used instruments to measure BC mass concentration have been compared during a two-years measurement campaign at the high altitude site Pic du Midi in the french Pyrenees. The agreement between a SP2, an AE33 and a Sunset measuring refractive BC (rBC), equivalent BC (eBC) and elemental carbon (EC), respectively, have been studied and the causes of variability investigated.

All techniques responded to seasonal variations of the atmospheric changes in BC levels and exhibited good correlation during the whole study period. This indicates that the different instruments quantified the same particle type, despite the fact that they are based on different physical principle. However the slopes varied between instrument pairs. The largest biases were observed for the AE33 $M_{eBC}$ values that were larger by a factor of around 2 than SP2 $M_{rBC}$ and Sunset $M_{EC}$ values. The SP2 and Sunset measurements appear to have the best agreement with an average bias by around 17 % between $M_{EC}$ and $M_{rBC}$. However large overestimation of $M_{EC}$ compared to $M_{rBC}$ and $M_{eBC}$ is observed when $M_{TC}$ was lower than 25 $\mu$gC cm$^{-2}$



of TC. Our analyses indicate a possible underestimation of $M_{PyrC}$ for some of these samples that could partly explain the positive bias in Sunset measurements. This threshold TC value is higher than the value of 10 $\mu$gC cm$^{-2}$ obtained during the

multiple-Sunset intercomparison study performed at a PBL background site in Italy (EMEP/CCC–Report 1/2018). We note that the aerosol types measured in this last intercomparison study may be very different to those studied here, and therefore this result could indicate a dependence of the lower quantification limit of the Sunset to the thermal properties of the sampled carbonaceous particles. This threshold is a real issue for remote sites where low aerosol concentrations prevail and raises the need for alternative measurement techniques at low TC loading.

The main source of bias in $M_{rBC}$ measurements is found to be the limited size detection range of the SP2, which do not allow the detection of all rBC particles. Sensitivity tests based on different fitting approaches varying in terms of time resolution and number of lognormal modes has been carried out. While most studies use a fit with a single mode and averaged over the entire campaign, this approach does not adequately reproduce the rBC size distribution observed at PDM. Our results indicate that considering the daily variation and multimodal shape of rBC size distribution is required in the fitting procedure for accurately

quantify $M_{rBC}$ at PDM.

The systematic positive bias in AE33 compared to SP2 and Sunset was attributed to the C×MAC values applied for the $M_{eBC}$ retrievals. The best agreement between $M_{eBC}$ and both $M_{rBC}$ and $M_{EC}$ was obtained when C×MAC values were around 1.9 to 2.3 times higher than those recommended by the manufacturer. C×MAC values were found to be seasonal-depend and strongly linked to the source of rBC (determined using $\Delta M_{rBC}/\Delta CO$ tracer). Another cause of bias in AE33 is found to be the sampling

of dust particles that causes a large overestimation of $M_{eBC}$ by up to a factor of 8. The incorrect C value applied during dust events may be the main cause of such a discrepancy.

Based on the results and specific issues presented above, this study points out some recommendations for improving the assessment of $M_{EC}$, $M_{rBC}$ and $M_{eBC}$ :

1. The low detection sensitivity in separating accurately OC and EC at low TC contents with the Sunset Analyser makes
the use of this instrument tricky at some remote and background measurement sites under low pollution conditions.

2. A special attention should be paid to the rBC procedure used to estimate the missing mass fraction of rBC not covered by the SP2 measurement range. The temporal resolution and the number of modes required to fit the rBC size distribution can vary greatly from one region to another and from one season to another.

3. We recommend to remove periods under strong dust events from the AE33 dataset, which could lead to a large overesti-
mation $M_{eBC}$.

4. If possible, the systematic deployment of an additional on-line instrument to measure absorption coefficient unaffected by filter artefacts would be very useful to constrain the correction factor C applied in AE33 retrievals.

*Data availability.* rBC, EC and eBC data are available upon request to the authors.



*Competing interests.* The authors declare that they have no conflict of interest.

*Acknowledgements.* This work received funding from the French national program LEFE/INSU and Météo-France. Observation data were collected at the Pyrenean Platform for Observation of the Atmosphere P2OA (http://p2oa.aero.obs-mip.fr) in the frame of the SNO-CLAP (National Observatory Service for CLouds and Aerosols Properties). P2OA facilities and staff are funded and supported by the University Paul Sabatier Toulouse 3, France, and CNRS (Centre National de la Recherche Scientifique). We especially thank the staff of the Pic du Midi platform (Observatoire Midi-Pyrénées) for their technical assistance. We acknowledge the SNO ICOS-France and ACTRIS-France for

supporting aerosol observations at PDM, data collection, processing and dissemination.



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
