# Peer review of "Figure S1: rBC size distribution observed and fitted with one (dashed black line) or three (solid black line) lognormal modes for 2019 and 2020 at the PDM grouped by seasons."

_EGUsphere, 2024_

## Author Comment (AC1)

**Answer to referees**

**Two-year intercomparison of three methods for measuring black carbon concentration at a high-altitude research station in Europe**

We thank the two reviewers for evaluating the manuscript and providing us constructive and useful comments.

Please find below reviewer comments in black and our responses in blue. The line numbers in the responses refer to the new version of the paper.

**Anonymous Referee #1**

Tinorua et al., 2024 "Two-year intercomparison of three methods for measuring black carbon concentration at a high-altitude research station in Europe" provides new findings on the uncertainties and specific artifacts encountered when using different techniques in determining the mass concentration of atmospheric black carbon. The manuscript is based on two years of atmospheric data and fits well with the scope of the journal. The text is well written and structured and derives rather consistent conclusions based on the analysis presented. This said, however, the analysis is rather superficial and presents mainly temporal variability of correlations and statistical uncertainty analysis. As such, the manuscript provides rather minor additions on top of the already published article by Tinorua et al., 2023 in ACP and does not evolve the analysis methodologies further towards the goals of this manuscript. The author should take advantage of the available size distribution data (existing based on Tinorua et al., 2023) and the measured aerosol optical properties (such as SSA and AAE) when evaluating the causes of the observed discrepancies in BC measurements. For example, when speculating on the ultrafine rBC particles from aviation (L342) or on the variability of the multiple scattering correction factor (P20-P21), these additional data could provide further insights for the underlying reasons behind the observations. Therefore, I would like to encourage the authors to incorporate into the analysis both the aerosol number size distribution and the aerosol single scattering albedo, as additional parameters to consider when different artifacts are evaluated.

REPLY:

Tinorua et al. (2024) is a manuscript on atmospheric processes occurring during 2019-2020 at the Pic du Midi. It describes the optical and microphysical properties of BC and shows the dependence of these properties with the boundary layer dynamics, the BC emission sources, transport pathways and chemical reactivity. The current paper is a technical manuscript that aims to highlight, quantify, and find the sources of the biases between the three most used methods to quantify BC mass concentrations. Thus, we do not believe that the current paper should be considered as an addition to Tinorua et al. (2024) but as a technical intercomparison study based on the same measurement campaign which could- but not only- be complementary to the ACP study.

We thank the reviewer for his relevant suggestion of exploring the aerosol particle size distribution to verify the presence of ultrafine particles from aviation, and thus explain the bias on $M_{rBC}$ measured by the SP2. We plotted the aerosol size distribution as a function of the

$M_{EC}/M_{rBC}$ ratio in Figure 7 (cf. Fig 7 below). This new Figure clearly shows a dominant presence of small particles below 20 nm when $M_{EC}/M_{rBC}$ ratio reveals a significant positive bias. This mode is completely absent when the bias on $M_{EC}/M_{rBC}$ ratio is negative or neutral.

A text on l. 355-359 has been added to describe Figure 7:

"To further investigate the role of small particles on the $M_{rBC}$ bias, the aerosol number size distributions grouped by $M_{EC}/M_{rBC}$ ratios has been plotted in Fig. S4 in the Supplement. The size distribution for which the highest ratios between $M_{EC}$ and $M_{rBC}$ has been observed clearly shows a contribution of small particles (<20 nm) up to 2 times higher than when the bias on the $M_{EC}/M_{rBC}$ ratio is negative or neutral, supporting that the $M_{rBC}$ underestimation compared to $M_{EC}$ is probably due to undetected small rBC-containing particles."

[Figure]

Figure 7: a) Number size distribution of aerosols measured by the SMPS, colored by $M_{EC}/M_{rBC}$ ranges of values. b) is the same as a) but normalised by the total aerosol number concentration. Vertical lines highlight geometrical diameter corresponding to the color of the $M_{EC}/M_{rBC}$ range.

Based on Tinorua et al., 2023, the measured aerosol SSA was rather high (>0.9). How does the CxMAC value depend on the aerosol optical properties and the aerosol particle size distribution? Do the mass correlations present additional dependence on them?

REPLY: Since the SSA is calculated with $\sigma_{abs}$, which is retrieved from $\sigma_{ATN}$ using the C value, we believe that investigating the MAC*C dependency on the SSA is not relevant. We can notice that Yus-Díez et al. (2021) were able to show an increasing C value with the SSA because they had two different measurements of $\sigma_{abs}$, which is not the case in our study.

In addition, I have some minor comments for the authors to consider, presented below.

Specific comments:

- Please, be specific when defining "MAC" – do you mean MAC of the aerosol or of the material black carbon? Especially in the introduction (lines 49-52) it is slightly confusing what is meant by MAC.

REPLY: The sentence in l. 48-50 has been modified as follows:

"Quantifying $M_{eBC}$ acquired by optical methods is challenging because it requires the assumption of a BC mass absorption cross section (MAC) value translating the absorption coefficient ($\sigma_{abs}$)."

- L54 Note that "optical method" could include also other than filter-based absorption measurements.

REPLY: This has been replaced in l. 54 by the term "filter-based optical methods".

- L 358-360 Consider simplifying and sharpening the key point of the sentence. For example, it seems rather intuitive that possibly "the SP2 missed the detection of a mode that is centered at lower diameter than the lower limit of detection of the SP2".

REPLY: The sentence in l. 368-369 has been modified as follows:

"(1) the extrapolation of the first mode peaking at ~ 100 nm is inaccurate for masses lower than 90 nm, which is the lower size detection limit of the SP2".

- L 367-368 Referring to a study by Wei et al., 2020, the MAC values provided here are now a bit different than in introduction, also specifying that the MAC is for BC (material?). Please double check this reference and the correct values.

REPLY: This mistake has been corrected in l. 377-378 as follows:

"Nonetheless a wide range of MAC of BC from 3.8 to 58 $m^2$ $g^{-1}$ at 880 nm has been reported from field and laboratory measurements (Wei et al., 2020)."

- L388 Please provide a bit more information on the model and how it was applied, e.g. for which altitude and what meteorological data were utilized.

REPLY: Some details about the Hysplit model has been added in Section 2.3, l. 205-209 as follows:

"The Hysplit model (Hybrid Single-Particle Lagrangian Integrated Trajectory, Stein et al., 2015) has been used to retrieve the precipitation event along the 72-h trajectory of air masses arriving at the measurement site. The model was initialised to the PDM altitude, using 3-hourly

atmospheric data of 1-degree spatial resolution from the Global Data Assimilation System (GDAS) of the National Centers for Environmental Prediction (NCEP)."

- L394 I would not recommend calling this observed behavior a "trend". L394 Explain what "measurement artifact" is suspected to explain the difference.

REPLY: The sentence of l. 404-406 has been modified as follows:

"The absence of correlation between $C \times MAC$ and $\Delta M_{rBC}/\Delta CO$ in spring may be due to a measurement artifact during this season, such as the dominant presence of dusts particles which can affect the C correction of the AE33 (cf. Fig. 10a. and associated text)."

References :

Stein, A. F., Draxler, R. R., Rolph, G. D., Stunder, B. J. B., Cohen, M. D., & Ngan, F. (2015). NOAA's HYSPLIT Atmospheric Transport and Dispersion Modeling System. *Bulletin of the American Meteorological Society*, *96*(12), 2059–2077. https://doi.org/10.1175/BAMS-D-14-00110.1

Tinorua, S., Denjean, C., Nabat, P., Bourrianne, T., Pont, V., Gheusi, F., & Leclerc, E. (2024). Higher absorption enhancement of black carbon in summer shown by 2-year measurements at the high-altitude mountain site of Pic du Midi Observatory in the French Pyrenees. *Atmospheric Chemistry and Physics*, *24*(3), 1801–1824. https://doi.org/10.5194/acp-24-1801-2024

Wei, X., Zhu, Y., Hu, J., Liu, C., Ge, X., Guo, S., Liu, D., Liao, H., & Wang, H. (2020). Recent Progress in Impacts of Mixing State on Optical Properties of Black Carbon Aerosol. *Current Pollution Reports*, *6*(4), 380–398. https://doi.org/10.1007/s40726-020-00158-0

Yus-Díez, J., Bernardoni, V., Močnik, G., Alastuey, A., Ciniglia, D., Ivančič, M., Querol, X., Perez, N., Reche, C., Rigler, M., Vecchi, R., Valentini, S., & Pandolfi, M. (2021). Determination of the multiple-scattering correction factor and its cross-sensitivity to scattering and wavelength dependence for different AE33 Aethalometer filter tapes: A multi-instrumental approach. *Atmospheric Measurement Techniques*, *14*(10), 6335–6355. https://doi.org/10.5194/amt-14-6335-2021

---

## Author Comment (AC2)

**Answer to referees**

**Two-year intercomparison of three methods for measuring black carbon concentration at a high-altitude research station in Europe**

We thank the two reviewers for evaluating the manuscript and providing us constructive and useful comments.

Please find below reviewer comments in black and our responses in blue. The line numbers in the responses refer to the new version of the paper.

**Anonymous Referee #2**

This study provides an intercomparison between three BC measurement techniques, using an aethalometer AE33, a thermal-optical analyzer Sunset and a single-particle soot photometer SP2. This work provides useful information as it evaluates the agreement between those three instruments with a 2-year dataset of measurements at a high-altitude research site and discusses possible reasons of biases. The text is well-structured and the results support well the conclusions. However, the following issues need to be addressed:

1. Table 2; first row & column "measurement uncertainty": There should be also some uncertainty in the mass calibration factor applied. Please add also a reference here.

REPLY: First row and column "measurement uncertainty" has been modified as follows :

"24.5% (quadratic sum of sampling flow , anisokinetic sampling errors and mass calibration factor errors), (Schwarz et al., 2006)"

2. Line 110: Can you please double-check that this is the correct LOT of fullerene soot and add the relationship that you used for the mass calibration?

REPLY: The lot of fullerene soot has been checked once again, and the number is FS12S011. The relationship has been added in l. 110 as follows:

"The calibration was performed using monodispersed fullerene soot (Alfa Aesar, lot #FS12S011) selected by a differential mobility analyzer and applying a second order polynomial fit."

3. Line 119: This needs to be rephrased. $R_{fit/meas}$ is the fraction of the estimated ambient rBC mass that is outside of the SP2 size detection limit.

REPLY: The sentence of l. 118-120 has been rephrased as follows:

"All these methods are based on fitting the measured rBC size distribution with lognormal distribution and estimating the ambient fraction of rBC mass outside the SP2 measurement range, hereafter referred as $R_{fit/meas}$ and calculated using Eq. 1:"

4. Line 122: The definitions of $M_{rBC, fit}$ and $M_{rBC, meas}$ need to be more clear.

REPLY: This has been modified in l. 122 as follows:

"where $M_{rBC,fit}$ is the fitted rBC size distribution between 1 and 1000 nm and $M_{rBC,meas}$ the measured rBC size distribution in the detection size range of the SP2."

5. Lines 134-136: Any uncertainties in BC mass calibration should be added.

REPLY: The sentence in l. 135-137 has been modified as follows:

"The resulting uncertainty on $M_{rBC}$ is estimated to be around 24.5 %, corresponding to the quadratic sum of the 20 % uncertainty on the mass calibration factor, the 10 % uncertainty for anisiokinetic sampling errors and the 10 % uncertainties on the flow calibration (Schwarz et al., 2006)."

6. Lines 152-155: The abbreviation used for Pyrolytic Carbon is not consistent in the text.

REPLY: This has been corrected.

7. Section 3.1:

The basis presented in the paper for the trimodal fit was the better representation of the size distribution within the SP2 range: "As a first conclusion, the trimodal curve generally better follows the measurements, and in particular for rBC diameter above 150 nm." However, such an improvement is always expected for fits with increasing numbers of parameters. To fully justify the reduced uncertainty associated with the more complex fit, it is important to have confidence that the physical basis for the additional parameters is justified. In this case, this reviewer wonders if there is a different explanation for the small structure quite consistently around 150 nm except in winter. Could the authors provide more information about the calibration of the incandescent detectors? Was just one detector used, or could there be a gain shift around this diameter? Was a linear relationship between peak height and rBC mass used, or a more complex relationship? Finally, was there any additional information allowing separation of these assumed modes?

Note that the manner in which the largest mode is being dealt with here (i.e. fitting a poorly constrained shoulder and including the resulting inferred mass) contrasts with previous approaches, in which the larger mode with some coarse-mode contributions was not presented in the context of the accumulation-mode rBC. Here, in Fig. 1, it's clear that the unimodal fit approximates the larger-particle mass contributions without significantly extrapolating to the coarse mode.

REPLY: A calibration curve based on a second order polynomial relationship has been calculated, as has been previously done by Taylor et al (2015). As it can be seen on Fig 1 below, the fitted calibration curve for the high gain (in orange) agrees (considering an uncertainty of 24.5 % on $M_{rBC}$) with the fitted calibration curve calculated with the low gain measurements

multiplied by a factor of 10 (in purple). No shift between the two gains has been noticed over the SP2 size detection range.

[Figure]

*Figure 1: Calibration curves of the SP2 detectors for low gain and high gain in blue and orange, respectively. . Dots represent the average calibration measurements and dotted lines shows the fitted measurements. The low gain multiplied by a factor of 10 has been added in purple to show the agreement with the high gain.*

As noticed by the reviewer, the rBC size distribution shape changed as a function of the season, and especially, the structure around 150 nm was absent in winter, meaning that this structure is not a systematic artifact affecting the SP2 gains or detectors. Such a multimodal lognormal distribution has previously been observed, e.g. in Fresno, where Cappa et al. (2019) fitted the rBC size distribution with four modes.

1. Line 207: Better change to "coarser mode" (also later in the text, e.g. line 258).

REPLY: This has been modified.

Line 218: Here $M_{rBC, fit}$ is not defined as in Equation 1. Please correct either the symbol or the definition.

However, I recommend when evaluating the different fitting approaches, to present changes on the ratio of the rBC mass under the whole fitted area to the rBC mass measured over the size range covered by the SP2. This will give a straightforward comparison between the mass correction factors that you had to apply for estimating the total (accumulation?) rBC mass and will be better connected to the $R_{fit/meas}$ that you discuss later in the text.

REPLY: The $M_{rBC,fit}$ / $M_{rBC,meas}$ ratio already shows the rBC mass under the whole fitted area (1 to 1000 nm ) over the rBC mass measured within the size range covered by the SP2. This has been clarified on l. 122 (cf. answer on comment # 4).

The definition of $M_{rBC,fit}$ in l . 225 has been modified to match the previous one as follows:

"Figure 2 shows the ratio between $M_{rBC}$ fitted between 1 and 1000 nm from and the one derived from the observation ($M_{rBC,fit}/M_{rBC,meas}$) over the $D_{rBC}$ range covered by the SP2 for the different fitting approaches throughout the campaign (Figure 2a) and the overall statistical results (Figure 2b)."

The legend of Fig. 2 has been corrected.

1. Line 228 and later in the text: The largest mode is around 400 nm (given the fitted peak at 377 nm).

REPLY: This has been corrected.

2. Line 236: The sentence "larger differences in rBC…" is better to be removed as this statement is also given in lines 239-240 with the right reference (i.e., Fig. 1).

REPLY: This sentence has been removed and the sentence in l. 244-246 has been modified as follows:

"Although the overall bias between $M_{rBC,meas}$ and $M_{rBC, fit}$ remained low ( < 2% on average over the 2-year campaign) regardless the approach chosen, it can be seen in Figure 1 that the largest differences between the 1-mode and the 3-modes fitting approaches occurs for $D_{rBC}$ < 90 nm."

3. Line 281: Should be "$M_{EC}$ vs $M_{eBC}$"

REPLY: This has been corrected.

4. Line 400: Refer to Figure10a

REPLY: This has been corrected.

References:

Cappa, C. D., Zhang, X., Russell, L. M., Collier, S., Lee, A. K. Y., Chen, C.-L., Betha, R., Chen, S., Liu, J., Price, D. J., Sanchez, K. J., McMeeking, G. R., Williams, L. R., Onasch, T. B., Worsnop, D. R., Abbatt, J., & Zhang, Q. (2019). Light Absorption by Ambient Black and Brown Carbon and its Dependence on Black Carbon Coating State for Two California, USA, Cities in Winter and Summer. *Journal of Geophysical Research: Atmospheres*, *124*(3), 1550–1577. https://doi.org/10.1029/2018JD029501

Schwarz, J. P., Gao, R. S., Fahey, D. W., Thomson, D. S., Watts, L. A., Wilson, J. C., Reeves, J. M., Darbeheshti, M., Baumgardner, D. G., Kok, G. L., Chung, S. H., Schulz, M., Hendricks, J., Lauer, A., Kärcher, B., Slowik, J. G., Rosenlof, K. H., Thompson, T. L., Langford, A. O., … Aikin, K. C. (2006). Single-particle measurements of midlatitude black carbon and light-scattering aerosols from the boundary layer to the lower stratosphere. *Journal of Geophysical Research: Atmospheres*, *111*(D16). https://doi.org/10.1029/2006JD007076

Taylor, J. W., Allan, J. D., Liu, D., Flynn, M., Weber, R., Zhang, X., Lefer, B. L., Grossberg, N., Flynn, J., & Coe, H. (2015). Assessment of the sensitivity of core / shell parameters derived using the single-particle soot photometer to density and refractive index. *Atmospheric Measurement Techniques*, *8*(4), 1701–1718. https://doi.org/10.5194/amt-8-1701-2015

---

## Author Response (AR2)

**Answer to referees**

**Two-year intercomparison of three methods for measuring black carbon concentration at a high-altitude research station in Europe**

Please find below reviewer comments in black and our responses in blue.

The authors have resolved most of the issues however, there are still some parts that need to be carefully revised:

1. L. 122 Please describe better MrBC, fit and MrBC, meas. For example, is MrBC,fit the total rBC mass derived by integrating the fitted size distributions over the diameter range 1-1000 nm?

2. L. 225 The reason of fitting a log-normal is to estimate accumulation-mode rBC mass that is outside the detection range of SP2. I would expect then, the total rBC mass derived from the fit to be always higher than the rBC mass that has been measured by SP2 and therefore the ratio MrBC,fit/MrBC,meas to be greater than 1. However, in l. 227 you state that "the unimodal fit tends to slightly underestimate MrBC by around 1.6% regardless of the selected averaging time". Looking at Fig. 1 and taking into account that MrBC, fit is indeed the integral total mass of rBC from the fitted lognormal over the diameter range 1-1000 nm, MrBC,fit/MrBC,meas cannot be less than 1. There is a confusion here because of the given definitions and this part needs still to be corrected.

REPLY:

We understand the confusion of the reviewer. To clarify that, we changed the notation of the different terms implying the rBC mass concentration throughout the whole manuscript, equations, and figures according to the editor's suggestion so that:

- MrBC,meas refers to the mass of rBC as reported directly from the SP2 measurements

- MrBC,corr refers to the mass of rBC corrected for the extrapolated mass using a time-dependent correction factor based on lognormal fits on the rBC size distributions

- MrBC,fit refers to the mass of rBC calculated from the rBC size distribution lognormal fits by integrating the fits from 1 nm to 1000 nm.

Besides, we have read in depth the manuscript and addressed the minor edits raised by the editor. In particular, the color scale in Fig. 4 has been modified in line with the ACP's Copernicus guidelines, and the legends of Fig.4 and Fig. 5 have been refined.